# AerialGait: Bridging Aerial and Ground Views for Gait Recognition

## ABSTRACT

In this work, we present AerialGait, a comprehensive dataset for aerial-ground gait recognition. This dataset comprises 82,454 sequences totaling over 10 million frames from 533 subjects, captured from both aerial and ground perspectives. To align with real-life scenarios of aerial and ground surveillance, we utilize a drone and a ground surveillance camera for data acquisition. The drone is operated at various speeds, directions, and altitudes. Meanwhile, we conduct data collection across five diverse surveillance sites to ensure a comprehensive simulation of real-world settings. AerialGait has several unique features: 1) The gait sequences exhibit significant variations in views, resolutions, and illumination across five distinct scenes. 2) It incorporates challenges of motion blur and frame discontinuity due to drone mobility. 3) The dataset reflects the domain gap caused by the view disparity between aerial and ground views, presenting a realistic challenge for drone-based gait recognition. Moreover, we perform a comprehensive analysis of existing gait recognition methods on AerialGait dataset and propose the Aerial-Ground Gait Network (AGG-Net). AGG-Net effectively learns discriminative features from aerial views by uncertainty learning and clusters features across aerial and ground views through prototype learning. Our model achieves state-of-the-art performance on both AerialGait and DroneGait datasets. The dataset and code will be made available upon acceptance.

## CCS CONCEPTS

• **Computing methodologies → Biometrics**; **Object identification**.

## KEYWORDS

Gait Recognition, Datasets, Aerial Views, Uncertainty Learning, Prototype Learning

## 1 INTRODUCTION

Gait recognition can be conducted at a distance without the cooperation of subjects. This characteristic establishes it as an essential tool for crime prevention, forensic identification, and social security. Integrating gait recognition within aerial surveillance systems introduces a novel paradigm in monitoring and security applications. The development of drone-based applications in both industrial and

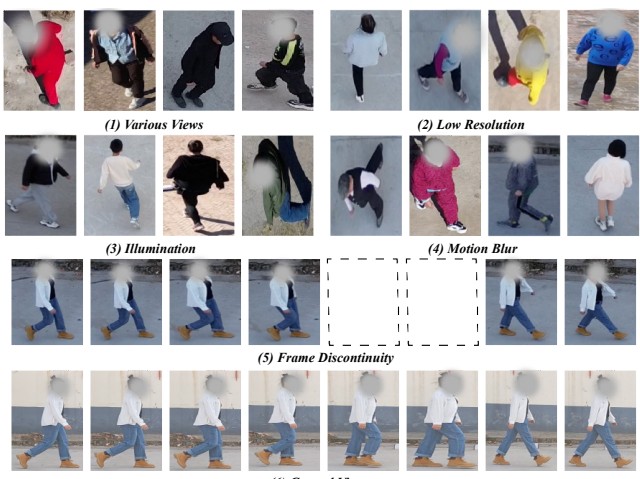

**Figure 1: Challenges in AerialGait: (1) Various views, (2) Low resolution, (3) Illumination, (4) Motion blur, and (5) Frame discontinuity. In the last row, we provide an example of ground view sequence.**

academic areas has solidified the role of drones for aerial surveillance. Compared to ground-based surveillance, aerial surveillance offers unparalleled advantages including mobility, the efficiency of tracking individuals, and the capability for both covert and overt surveillance [17].

However, the research on drone-based gait recognition is limited, and the existing gait recognition datasets are collected by fixed surveillance cameras under static scenes [14, 20, 35, 39]. Although recent research has introduced a drone-based gait recognition dataset [12], it captures gait sequences by a stationary aerial platform. This approach does not leverage the inherent mobility and flexible target tracking capabilities of drones, which does not align with real-world application scenarios. Furthermore, this dataset is constrained by a limited number of subjects and fixed environment. Consequently, there is a clear need for a more diverse and extensive dataset that contains both aerial and ground views for gait recognition.

In this work, we introduce AerialGait, a comprehensive large-scale dataset for aerial-ground gait recognition. AerialGait contains 82,454 sequences with 10,259,669 frames from 533 different identities. These subjects are captured from both aerial and ground views. AerialGait is collected using a ground camera and a DJI Mavic 3 drone. The operation of the drone encompasses a diverse range of speeds, directions, and altitudes, introducing a variety of views, resolutions, and motion blur effects to accurately align with real-world aerial-ground surveillance scenarios. To enhance the dataset's diversity and applicability, subjects are filmed across five

**Table 1: Comparison of AerialGait with other gait recognition datasets.**

| Dataset | Year | Subject # | Seq # | Data Type | Aerial View | Various Scenes | Moving Camera |
|---------|------|-----------|-------|-----------|-------------|----------------|---------------|
| CASIA-B [29] | 2006 | 124 | 13,640 | RGB, Silhouette | ✗ | ✗ | ✗ |
| OU-MVLP [22] | 2018 | 10,307 | 288,596 | Silhouette | ✗ | ✗ | ✗ |
| GREW [39] | 2021 | 26,345 | 128,671 | Silhouette, 2D/3D Pose, Flow | ✗ | ✓ | ✗ |
| Gait3D [35] | 2022 | 4,000 | 25,309 | Silhouette, 2D/3D Pose, 3D Mesh | ✗ | ✓ | ✗ |
| CASIA-E [21] | 2022 | 1,014 | 778,752 | Silhouette | ✗ | ✓ | ✗ |
| SUSTech1K [20] | 2023 | 1050 | 25,279 | RGB, Silhouette, 3D Point Cloud | ✗ | ✓ | ✗ |
| CCPG [14] | 2023 | 200 | 16,566 | RGB, Silhouette | ✗ | ✗ | ✗ |
| Gait3D-Parsing [36] | 2023 | 4,000 | 25,309 | Parsing | ✗ | ✓ | ✗ |
| DroneGait [12] | 2023 | 96 | 22,718 | Silhouette, 2D/3D Pose, Flow, Mesh | ✓ | ✗ | ✗ |
| AerialGait | - | 533 | 82,454 | Silhouette, 2D/3D Pose, Parsing | ✓ | ✓ | ✓ |

distinct locations, including crossroads, rural paths, and stadiums, ensuring various background scenarios.

The comparison of AerialGait and other prominent gait datasets is depicted in Table 1. This comparison highlights that only the DroneGait [12] dataset and AerialGait include aerial view sequences. However, DroneGait employs a stationary aerial platform, foregoing the mobility of drones and thus diverging from realistic deployment scenarios. To better approximate real-world applications, where drones are moving under different environments for individual identification and continuous tracking. We use a mobile drone to capture gait sequences across various scenes, thereby enriching the dataset with diverse background scenarios. Furthermore, the AerialGait dataset not only equals but surpasses the recent datasets like CCPG [14], SUSTech1K [20], and DroneGait [12] in terms of sequence numbers.

Under real-world scenarios, aerial-ground gait recognition faces several unique challenges, as shown in Figure 1. These challenges include **significant variations in views, resolutions, and illumination.** The differences in heights and directions of drone flight introduce a variety of views. For instance, high-altitude flights capture top-down views, which may obscure specific gait features, while lower flights offer detailed side views that better reveal walking patterns. Furthermore, the resolution is influenced by the distance between the drone and the subject, resulting in disparate resolutions among different subjects. Additionally, collecting gait data across diverse temporal and environmental conditions introduces challenges such as variations in illumination and background.

In addition, **motion blur and frame discontinuity** stem from the movement of both drone and subject, particularly when the drone operates at varying speeds (1m/s-15m/s) and altitudes (3m-20m). The moving drone makes captured images blurry and sometimes causes the target to move out of the drone's field of view, leading to discontinuous frames. Motion blur deteriorates the quality of RGB images, and impacts downstream methods that rely on RGB images to generate gait data, such as segmentation and pose estimation [19]. Frame discontinuity disrupts the sequence order of captured frames, which is critical for the analysis of temporal movement patterns.

Besides, the **domain gap caused by view disparity between aerial and ground views** brings significant challenges when conducting aerial-ground gait recognition. Models are required to learn view-invariant features to effectively match the same identity across aerial and ground views. These challenges highlight the complexity of aerial-ground gait recognition and emphasize the necessity for approaches to effectively integrate aerial and ground views.

In response to these complex challenges, we present the Aerial-Ground Gait Network (AGG-Net) with two main modules: Gait-Oriented Uncertainty Learning module and Aerial-Ground Prototype Learning module. **1)** Aerial surveillance encounters challenges such as variations in views, resolutions, illumination, and motion blur. To address these challenges, we propose the Gait-Oriented Uncertainty Learning module. This module introduces uncertainty at both the input and feature levels, designed to enhance the model's generalization ability in the presence of significant covariance in the data. **2)** Given the domain gap between aerial and ground views, we utilize the Aerial-Ground Prototype Learning module to mitigate the domain gap between aerial and ground views. Specifically the Aerial-Ground Prototype Learning module extracts view-specific features and updates the prototypes based on their corresponding identity labels, thereby aligning the feature distribution of aerial and ground views.

To summarize, the main contributions of our work are as follows:

- We construct a large-scale aerial-ground gait recognition dataset named AerialGait for the application of drone-based gait recognition. This dataset has the following unique characteristics: 1) Significant variations in views, resolutions, and illumination. 2) Motion blur and frame discontinuity. 3) Domain gap between aerial and ground views.
- Based on these challenges, we present the Aerial-Ground Gait Network (AGG-Net). The proposed AGG-Net is designed to effectively learn generalizable features from aerial perspectives and to bridge the discrepancy between aerial and ground views.
- The proposed method achieves state-of-the-art performance on both the AerialGait and the DroneGait datasets. Additionally, the ablation study shows the contribution and efficacy of the individual modules comprising the AGG-Net.

## 2 RELATED WORK

### 2.1 Ground View Gait Recognition

Recent advancements in ground view gait recognition have expanded from indoor environments to outdoor settings, which are more closely with real-world applications. Early research predominantly relied on in-the-lab datasets like CASIA-B [29] and OU-MVLP [22], which present limited variations in views, resolutions, and illumination, thereby restricting their real-world applicability. In response to these limitations, in-the-wild datasets such as Gait3D [35] and GREW [39] are introduced, which utilize surveillance cameras in supermarkets and streets, respectively. These datasets incorporate real-world complexities such as occlusions and varying lighting conditions, providing a more comprehensive evaluation framework for gait models. Furthermore, the SUSTech1K [20] dataset introduces lidar point cloud as a new data modality for gait recognition. Gait3D-Parsing [36], building on the Gait3D [35] dataset, incorporates human parsing as a new modality, significantly enhancing the performance of silhouette-based methods. Additionally, CCPG dataset [14] is constructed to demonstrate the robustness of gait recognition in cloth-changing condition, highlighting its advantages over traditional person re-identification methods.

Gait recognition methods can be divided into two categories based on the input modality: pose-based [6, 23, 31] and appearance-based methods [1, 4, 15, 25–27]. Additionally, recent efforts aim to combine multiple modalities [3, 20, 36] for a more comprehensive gait analysis. Pose-based approaches often rely on either 2D or 3D pose data. For instance, GaitTR [31] employs self-attention mechanism to investigate spatial correlations. GPGait [6] introduces a Human-Oriented Transformation, aiming to improve the method's generalizability across different datasets. Appearance-based methods primarily utilize silhouette as the input. GaitGL [15] focuses on both global and local features and proposes local temporal aggregation to integrate temporal information. GaitSet [1] regards the gait sequence as an unordered set and extracts set-based features. GaitBase [4] employs a ResNet-like backbone and simplifies GaitSet's network architecture, establishing a strong baseline across various benchmarks. LandmarkGait [27] generates landmarks from silhouettes, thereby constructing specific and comprehensive local representations of body parts through landmarks.

### 2.2 Aerial View Human Identification

With the rapid advancement of drone technology, numerous aerial view human identification datasets have emerged [7, 10, 11, 13, 16, 32, 33] to support research in aerial visual tasks. These datasets typically exhibit more complex intra-class variations, including differences in views and poses, compared to conventional visual datasets. For instance, PRAI-1581 [33], released in 2019, includes 39,461 aerial view images of 1,581 subjects. UAV-human [13], introduced in 2021, comprises 41,290 images of 1,144 individuals, captured by drones at altitudes ranging from 2 to 8 meters. Additionally, G2APS [32], the first to construct a large-scale ground-to-aerial person search benchmark dataset, contains 31,770 images of 2,644 identities, captured using both drones and ground surveillance cameras.

Drone-based gait recognition has also advanced in recent years, with DroneGait [12] is the only publicly available dataset. However,

DroneGait is collected using two stationary drones, which does not leverage the potential mobility of drones. This approach restricts the dataset's applicability in real-world scenarios, leading to limited variability in poses and views. Furthermore, the DroneGait dataset encompasses data from merely 96 subjects, further limiting its diversity. Alongside the dataset, a novel technique named Vertical Distillation [12] is introduced, aimed at refining aerial view features into a more distinctive distribution.

## 3 DATASET

The AerialGait dataset is collected by a ground camera and a drone, providing multi-view data in various outdoor environment. It contains 82,454 sequences with 10,259,669 frames from 533 different subjects. Additionally, it incorporates covariates such as carrying bags and changing clothes. The AerialGait dataset provides four types of data: silhouette, 2D/3D pose, and human parsing.

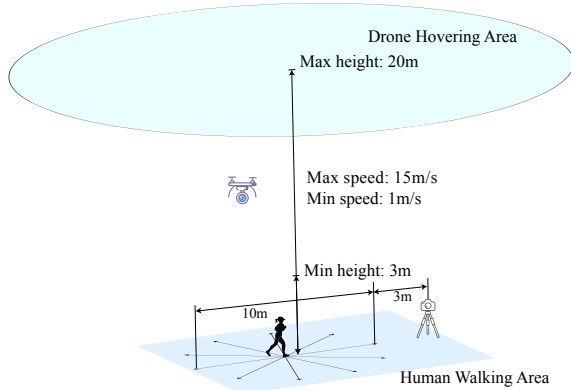

**Figure 2: The collection diagram of AerialGait. The drone's flight path is random, and subjects are instructed to walk in 10 different directions.**

### 3.1 Data Acquisition

The collection of AerialGait lasts for over two weeks across five distinct environments, ensuring a diversity of lighting conditions and background variations. Data collection is conducted using a monocular camera for ground views and a DJI Mavic 3 drone for aerial perspectives. The collection diagram is shown in Figure 2. To ensure a comprehensive simulation of real-world applications, subjects are instructed to walk in 10 different directions, while the drone is operated to hover and move in different directions. The drone's altitude is manually adjusted between 3m and 20m, with a random flight path. Flight speeds varied from 1m/s to 15m/s to replicate scenarios where drones search for targets. Monocular camera is positioned at a height of 1m. Both the drone and the monocular camera are set to a resolution of 1920×1080, recording at 30 frames per second (FPS). We refer to the setting with the CASIA-B [29] dataset, covariates of carrying bags and changing clothes are incorporated. Ideally, each subject contributes a total of 80 gait sequences, formulated as = $(4(normal\ walking) + 2(carrying\ bags) + 2(cloth\ changing)) \times 10(views)$.

## 3.2 Data Processing

We process the captured videos using human tracking methods [34], then manually remove irrelevant individuals. Finally, we construct a dataset comprising 82,454 sequences with a total of over 10 million frames from 533 subjects. To accommodate various gait recognition models, the dataset includes silhouette, 2D/3D pose, and human parsing result. Examples of different data types are shown in Figure 3. Human segmentation is performed using the HumanSeg model [2], human parsing are generated with U2-Net [18], 2D pose with ViTPose [28], and 3D pose with MotionBert [38].

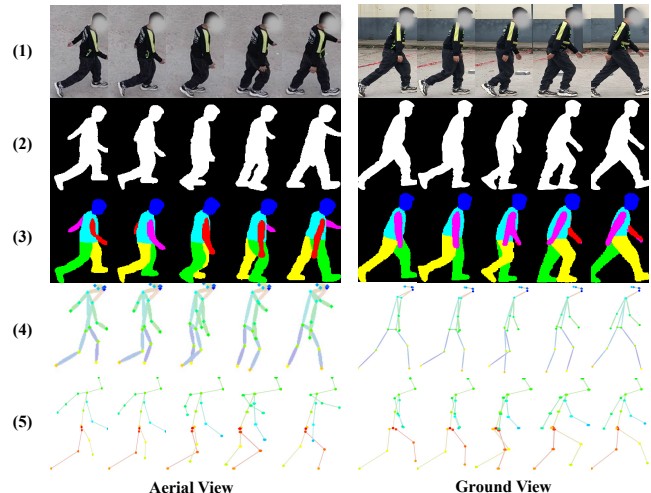

**Figure 3: Data types of AerialGait: (1) RGB, (2) Silhouette, (3) Human Parsing, (4) 2D Pose, (5) 3D Pose.**

## 3.3 Dataset Statistics

**Resolution Diversity.** The resolution distribution of aerial and ground views is illustrated in Figure 4 (a). **1)** Aerial views generally have a significantly lower average resolution compared to ground views. This discrepancy is primarily due to the greater distance of drones compared to ground-based cameras. **2)** The minimum resolution of ground view images is approximately 50,000 pixels, ensuring a basic data quality. However, the resolution of aerial views can be drastically lower due to considerable variations in flight altitude. This severe fluctuation significantly impacts the quality of gait data captured from aerial views.

**Sequence Length Diversity.** The distribution of sequence lengths for both ground and aerial views is depicted in Figure 4 (b). **1)** Both aerial and ground views exhibit a wide distribution. This phenomenon is attributed to the diverse collection environments. For example, subjects walking through crowded areas tend to walk slower compared to those in open areas, leading to variations in sequence lengths. **2)** The average sequence length of aerial views is significantly less than that of ground views. This discrepancy arises from drone mobility, when a drone flies too high or too fast, the tracking model will lose the target occasionally, resulting in fewer frames compared to ground views.

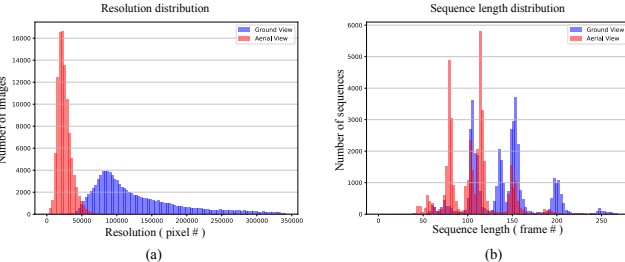

**Figure 4: Data statistics of AerialGait: (a) The resolution distribution of AerialGait. (b) The sequence length distribution of AerialGait.**

## 3.4 Privacy Statement

We are committed to ensuring the privacy and ethical of all subjects in our dataset. Each subject is openly recruited and participates voluntarily. Additionally, they are fully informed about the purpose and use of the dataset. Before data collection, we sign a data collection agreement with each subject, guaranteeing that their information will be only used for academic research purposes. Additionally, every researcher who wants to use the dataset is required to sign a dataset usage agreement, ensuring that the dataset will be only used for research purposes.

## 4 OUR APPROACH

In this section, we first describe the overview of our method in Section 4.1, and then introduce the proposed two well-designed modules, *i.e.*, Gait-Oriented Uncertainty Learning in Section 4.2 and Aerial-Ground Prototype Learning in Section 4.3. Finally, we introduce the overall loss function in Section 4.4.

## 4.1 Overview

Previously, we introduce three primary challenges in aerial-ground gait recognition, including 1) significant variations in views, resolutions, and illumination; 2) motion blur and frame discontinuity; and 3) domain gap between aerial and ground views. To address these challenges, we employ three distinct strategies.

Firstly, we tackle **frame discontinuity**, as the frames in aerial views are not always continuous. So we utilize a set-based 2D backbone that processes unordered sets of frames, which is more robust to the frame discontinuity. Secondly, the **significant variations in views, resolutions, illumination, and motion blur** hinder the model's ability to capture identity-specific information. Based on this premise, we propose Gait-Oriented Uncertainty Learning module, which comprises *Silhouette-Oriented Uncertainty Learning (SOUL)* and *Feature-Oriented Uncertainty Learning (FOUL)* submodules. The Gait-Oriented Uncertainty Learning module introduces uncertainty at both the input and feature levels, aiming to enhance the model's generalization ability when faced with input data exhibiting significant covariance. Lastly, to address the **domain gap between aerial and ground views**, we propose Aerial-Ground Prototype Learning module. Here, "prototype" refers to the representative examples in the dataset. By aligning the prototype

distribution of aerial and ground views, we bridge the gap between them effectively.

The architecture of AGG-Net is shown in Figure 5. Our model simultaneously samples both aerial and ground views of the same subject as input. Then the SOUL module generates diverse silhouettes based on the input, Subsequently, these generated silhouettes are fed into the feature extraction backbone. Following this, the FOUL module then converts these features into Gaussian distributions with distinct means and variances. Temporal and Horizontal Pooling are applied to aggregate the features. Then the pooled features are used to compute ID Loss through fully connected layers. Finally, Aerial-Ground Prototype Learning module updates the view-specific prototypes by the pooled features and calculates the distribution discrepancy between these prototypes.

## 4.2 Gait-Oriented Uncertainty Learning

*4.2.1 Silhouette-Oriented Uncertainty Learning.* In aerial view gait sequences, the large view variation often results in significant occlusion of the human body, particularly affecting the visibility of discriminative regions such as the legs and feet. Additionally, the movement of the drone introduces blur at the edges of silhouettes. To address these challenges, we propose the SOUL module, the detailed procedure is shown in Figure 5 (a).

Firstly, we randomly select a structuring element to construct a kernel by different types such as cross, ellipse and rect. The variability in structuring elements facilitates diverse transformations of the original image, introducing uncertainty in the morphological processing. Then, we randomly select a rectangular region within the silhouette based on a predefined range of area ratios ($s_{max}/s_{min}$) and aspect ratios $r$. Finally, we randomly apply either Dilation or Erosion to the selected region. Dilation expands the foreground regions of the silhouette by setting the pixels of the structuring element to the maximum value. Erosion, on the other hand, shrinks the foreground regions by removing pixels from edges of the silhouette, thereby refining the silhouette's shape and potentially separating connected foreground regions.

Compared with Random Erasing [37], our SOUL module focuses on making subtle changes to the human shape rather than assigning the whole selected region to zero. This approach prevents the model from overemphasizing specific regions and enhances its robustness when dealing with silhouettes of ambiguous shapes.

*4.2.2 Feature-Oriented Uncertainty Learning.* In the task of aerial-ground gait recognition, gait sequences captured from aerial views exhibit higher ambiguity compared to those from ground views. To address this issue, we propose the FOUL module, which is designed to capture the inherent noise in the data by modeling the extracted features as Gaussian distributions. The mean of these distributions serves as the typical feature vector for ID matching, while the variance quantifies feature uncertainty, with noisy data typically exhibiting larger variances. During training, feature vectors are sampled from these Gaussian distributions, thereby enhancing the model's generalization ability to tackle the various challenges presented by aerial views.

Specifically, as shown in Figure 5 (b), we modify the final block of the feature extraction backbone into two separate branches: a mean branch, denoted as $f_\mu(\cdot)$, and a variance branch, denoted as

$f_\sigma(\cdot)$. The i-th gait feature in the training batch, processed by the feature extraction backbone, is denoted as $e_i$. Then the i-th mean vector $\mu_i = f_\mu(e_i)$ and variance vector $\sigma_i = f_\sigma(e_i)$ are generated by two separate branches. The generated variance vector can be treated as a measure of uncertainty for the i-th gait sequence. Thus, the original feature representation is transformed into a Gaussian distribution, represented by $z_i \sim \mathcal{N}(\mu_i, \sigma_i)$.

However, since $z_i$ is sampled randomly based on mean vector $\mu_i$ and variance vector $\sigma_i$, the gradient will not propagate back to the preceding layers. To address this issue, the reparameterization trick [30] is employed. It generates a sample $\epsilon$ from a standard Gaussian distribution where $\epsilon \sim \mathcal{N}(0, I)$. Then, the sampled feature representation is computed as $z_i' = \mu_i + \epsilon \sigma_i$, this method effectively separates the random component from the trainable parameters, allowing the gradient to backpropagate through the network.

To regulate the variance produced by the variance branch and prevent the trivial solution of variance decreasing to zero, we compute the L2 norm of the variance and a margin $\tau$ is introduced to ensure that the variance remains within a reasonable range. The uncertainty-constrained loss $L_{uc}$ is formulated as:

$$L_{uc} = max(0, \ \tau - \sum_{i=1}^{B} ||\sigma_i||_2) \tag{1}$$

Here, $B$ represents batch size and calculated as $B = b_1 \cdot b_2$, where $b_1$ denotes the number of identities in a training batch, and $b_2$ represents the sequence number of each identity. Equation 1 effectively ensures that the variance within a batch remains within an appropriate range.

During our experiments, we observe that only a single channel in $\sigma_i$ is active, contributing to the uncertainty constrained loss. Interestingly, the corresponding channel in $\mu_i$ is consistently zero during training, indicating that the network ignore the information from the active channel in $\sigma_i$, This phenomenon suggests a significant channel bias, as $\sigma_i$ appear to have no impact on the sampled feature $z_i'$. To address this channel bias, we implement layer normalization across the channel dimension for both the mean and variance vectors. Consequently, the final sampled representation, $z_i''$, is given by:

$$z_i'' = LN_1(\mu_i) + \epsilon LN_2(\sigma_i), \ \epsilon \sim \mathcal{N}(0, I) \tag{2}$$

Subsequently, the generated $\mu_i$ and $z_i''$ are input into two fully connected layers with shared weights. The ID loss is calculated as follows:

$$
\begin{aligned}
L_{id} &= \sum_{i=1}^{B} (Tri(\mu_i) + CE(\mu_i)) \\
L_{s-id} &= \sum_{i=1}^{B} (Tri(z_i'') + CE(z_i''))
\end{aligned} \tag{3}
$$

where $Tri(\cdot)$ and $CE(\cdot)$ denote the triplet loss and cross-entropy loss, $B$ represent the batch size. By projecting both the original feature and the sampled feature into different ID distributions, we enhance the model's robustness against the challenges posed by diverse views.

## 4.3 Aerial-Ground Prototype Learning

In the task of aerial-ground gait recognition, there is a notable domain gap between aerial and ground views. Ground views typically provide abundant gait information due to clear visibility of the human body and moderate camera-subject distance. In contrast, aerial

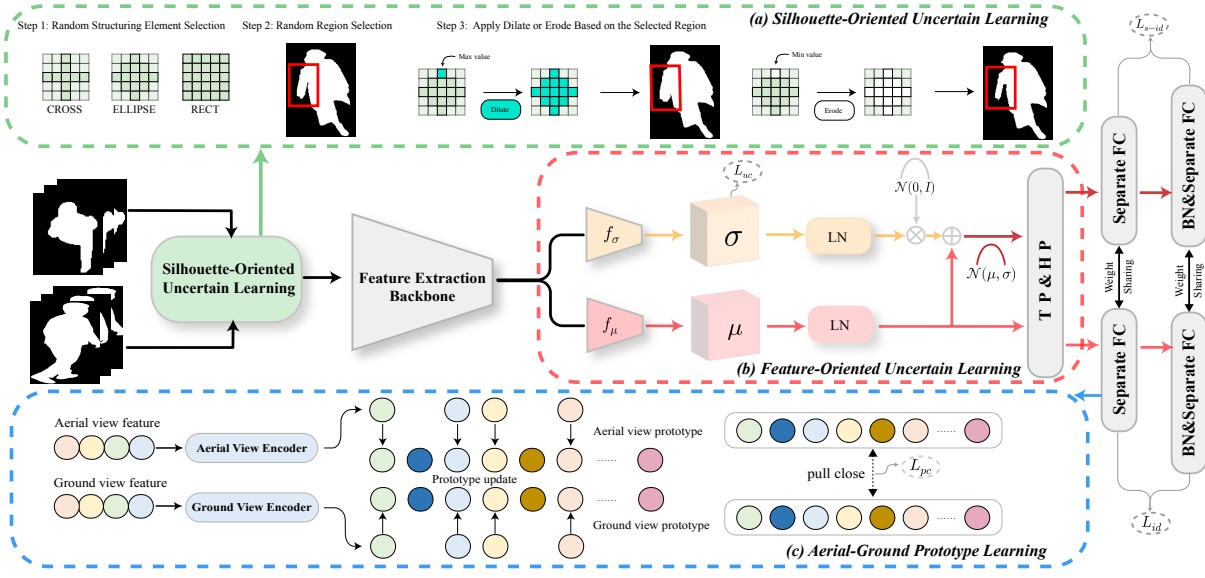

**Figure 5: The overview of the Aerial-Ground Gait Network (AGG-Net)**

views often faces challenges such as various views and motion blur, complicating the visibility and analysis of gait data. Given that these two perspectives share the same IDs but exhibit disparate characteristics, we introduce an Aerial-Ground Prototype Learning module. The prototype can be treated as the most representative examples in the dataset. By aligning the prototype distribution of aerial and ground views, we aim to alleviate the gap between them.

Specifically, as shown in Figure 5 (c). The prototypes of aerial and ground views are denoted as $P^a$ and $P^g \in N \times C \times D$. $N$ represents the number of identities in the training set, $C$ denotes the channel dimension, and $D$ indicates the part dimension. During the initial training stage, the prototypes are initialized as zero vectors. The features of aerial and ground views are denoted as $T^a$ and $T^g \in B \times C \times D$, where $B$ represent the batch size.

The momentum update strategy, inspired by the MoCo framework [8], is employed to update the prototypes with the features at each iteration. This strategy is defined as follows:

$$[p_i^v]^l = \begin{cases} \lambda Encoder^v([t_k^v]^l) + (1-\lambda)[p_i^v]^{l-1} & k = i \\ [p_i^v]^{l-1} & k \neq i \end{cases} \quad (4)$$

$$s.t.\ v \in \{a, g\}$$

where $[p_i^v]^l$ represents the $l$-th iteration prototype for identity $i$ and view $v$, where $v = a$ for aerial views and $v = g$ for ground views. Additionally, $[t_k^v]^l$ is the feature of the identity $k$. The feature is processed through a view-specific encoder $Encoder^v$ that incorporates a batch normalization layer and a fully connected layer. Then we iterate the features in the training batch. When $k = i$, the i-th prototype is updated based on $[t_k^v]^l$ iteratively. Parameter $\lambda$ is the update ratio. The momentum update strategy facilitates the incorporation of view-specific information and retains historical features, thus enabling synchronized clustering of prototypes.

Base on the view-specific prototypes, we calculate the prototype clustering loss ($L_{pc}$) during training. The loss is formulated as:

$$L_{pc} = mmd(P^a, P^g) = \phi(P^a, P^a) + \phi(P^g, P^g) - 2 \times \phi(P^a, P^g) \quad (5)$$

Here, the Maximum Mean Discrepancy (MMD) loss [9] is employed to minimize the distribution differences between aerial and ground views, where $\phi$ denotes the kernel function used to estimate the distribution difference. In our experiments, we utilize the Gaussian kernel function, which is detailed as follows:

$$\phi(P^a, P^g) = \frac{1}{N} \sum_{i=1}^{N} exp(-\frac{||p_i^a - p_i^g||^2}{2\xi^2}) \quad (6)$$

where $\xi$ is the width of the Gaussian Kernel. The MMD Loss assesses the distribution bias between aerial and ground views, effectively aligning the feature distributions of aerial and ground views.

### 4.4 Overall Loss Function

The basic loss in our framework are the ID loss $L_{id}$ which consists of standard triplet loss and CE loss [4]. In addition, we integrate the uncertainty constrained loss ($L_{uc}$), sampled id loss ($L_{s-id}$) from the FOUL module, and the prototype clustering loss ($L_{pc}$) from Aerial-Ground Prototype Learning module. Thus, the overall loss function is summarized as follows:

$$L_{total} = L_{id} + \alpha L_{s-id} + \beta L_{uc} + \gamma L_{pc} \quad (7)$$

where $\alpha$, $\beta$, and $\gamma$ are hyperparameters employed to balance the contribution of each loss during training.

## 5 EXPERIMENTS

### 5.1 Experimental Setup and Implementation Details

**Datasets.** We conduct experiments on the proposed AerialGait dataset and the DroneGait dataset [12].

**Table 2: Performance comparison on AerialGait and DroneGait.**

| Methods | Ref. | Modality | AerialGait | | | | DroneGait | | | |
|---------|------|----------|-----------|------|------|------|-----------|------|------|------|
| | | | A -> A | G -> G | G -> A | A -> G | A -> A | G -> G | G -> A | A -> G |
| GaitGraph2 [23] | CVPR22 | Skeleton | 24.62 | 32.11 | 16.49 | 18.05 | 20.50 | 36.48 | 11.24 | 12.44 |
| GaitTR [31] | ESWA23 | | 45.18 | 53.55 | 25.96 | 26.87 | 23.20 | 44.41 | 11.30 | 10.50 |
| GPGait [6] | ICCV23 | | 55.51 | 66.92 | 38.77 | 49.22 | 40.80 | 70.66 | 21.75 | 21.23 |
| GaitSet [1] | PAMI21 | Silhouette | 75.48 | 94.61 | 64.48 | 66.21 | 47.87 | 86.92 | 44.37 | 37.62 |
| GaitPart [5] | CVPR20 | | 57.41 | 85.81 | 41.74 | 41.80 | 41.81 | 87.10 | 39.31 | 30.01 |
| GaitGL [15] | ICCV21 | | 69.59 | 94.26 | 57.44 | 59.36 | 53.28 | 92.51 | 42.69 | 33.81 |
| V-Distill [12] | TMM23 | | 80.24 | 45.22 | 3.10 | 4.90 | 64.30 | 49.10 | 5.40 | 8.40 |
| GaitBase [4] | CVPR23 | | 81.22 | 96.51 | 71.64 | 74.83 | 65.75 | 91.59 | 50.06 | 44.38 |
| AGG-Net(ours) | - | | **84.92** | **97.32** | **76.00** | **80.35** | **69.69** | **92.33** | **54.44** | **48.97** |

**AerialGait** has 533 identities with 82,454 sequences. In our experimental setup, we randomly select 100 identities for training, while the remaining 433 subjects are set for testing. Specifically, during the testing phase, the first two sequences of normal walking are assigned to gallery, and the remaining sequences are utilized as probe.

**DroneGait** consists of 96 identities, with a total of 22,718 sequences. Follow the setting of the original paper [12], the first 48 identities are selected for training and the remaining 48 identities for testing. In the testing phase, the first two sequences of normal walking are assigned to the gallery, and the subsequent sequences are utilized as the probe. The DroneGait dataset is divided into three subsets, defined by their vertical viewing angles: $0°$, $30°$-$60°$, and $60°$-$80°$. For our experiments, we classify the $0°$ subset as ground views and the $60°$-$80°$ subset as aerial views.

**Evaluation Protocols** Our experiments are based on four evaluation protocols: Aerial to Aerial (A->A), Ground to Ground (G->G), Ground to Aerial (G->A), and Aerial to Ground (A->G). The first term in each pair represents the probe view, and the second term represents the gallery view. We compute the average Rank-1 accuracy under conditions of normal walking, carrying bags, and changing clothes. Additionally, we exclude the results when the subject's walking direction is the same in both the probe and gallery.

**Implementation Details.** We implement our model based on OpenGait [4], and all experiments are conducted on 8 NVIDIA TITAN V GPUs. The batch size is denoted as $[b_1, b_2]$, where $b_1$ represents the selected ID number, and $b_2$ represents the sequence number for each ID. In our experiments, we apply a batch size of [8,8] for both the AerialGait and DroneGait datasets, where each ID has 4 sequences from the aerial view and 4 sequences from the ground view. The network structure of the feature extraction backbone, along with the mean branch and variance branch in the Feature-Oriented Uncertainty module, is detailed in the Supplemental Materials. Here we provide the setup of hyperparameters in our experiments: **1)** The probability of applying the SOUL module is set to 0.5. The max/min area ratio range $s_{max}/s_{min}$ is set to 0.02 and 0.4, respectively, while the area aspect ratio range $r$ is set to 0.2. **2)** In the FOUL module, the margin $\tau$ for the uncertainty constrained loss is set at $4 \times 10^4$. **3)** In the Aerial-Ground Prototype Learning module, the updating ratio $\lambda$ of the prototype is set to 0.1. The width of the gaussian kernel $\xi$ is set to 1. **4)** The loss weights $\alpha$, $\beta$, and $\gamma$ are set to $1 \times 10^{-4}$, $1 \times 10^{-3}$, and $1 \times 10^{-1}$, respectively.

## 5.2 Experimental Results on AerialGait and DroneGait

This section presents a comprehensive analysis to evaluate the performance of gait recognition models on the AerialGait and DroneGait datasets. The detailed results is shown in Table 2. We compare our approach with three skeleton-based methods: GaitGraph2 [23], GaitTR [31], and GPGait [6], and five silhouette-based models: GaitSet [1], GaitPart [5], GaitGL [15], Vertical Distillation [12], and GaitBase [4].

*5.2.1 Comparison with State-of-the-art Models.* Among these models, our proposed AGG-Net surpasses existing approaches across four distinct evaluation settings on both the AerialGait and Drone-Gait datasets. **1)** In the Aerial to Aerial protocol, AGG-Net achieves 84.92% accuracy on the AerialGait dataset and 69.69% on the Drone-Gait dataset. Our model introduces uncertainty at data and feature levels, effectively capturing identity-related features and enhancing the model's robustness to noisy data. **2)** The model shows strong performance in the Aerial to Ground and Ground to Aerial protocols. On the AerialGait dataset, AGG-Net reaches a rank-1 accuracy of 80.35% in the Aerial to Ground protocol, outperforming Gait-Base [4] by a notable margin of 5.52%. Moreover, in the Ground to Aerial protocol of the DroneGait dataset, our method exceeds the second-best result by 4.49%. Our approach effectively align the distribution between aerial and ground views, thereby achieving superior performance in the four protocols.

*5.2.2 Comparison with Different Evaluation Protocols.* In the four evaluation protocols, the Ground-to-Ground setting yields the best results, reflecting the effectiveness of current gait recognition models under ground-based conditions. In contrast, the performance in Aerial-to-Aerial protocol shows a significant decline. It is largely due to challenges such as significant variations in viewpoint and resolution. Moreover, the Aerial-to-Ground and Ground-to-Aerial settings are further complicated by the domain gap between aerial and ground views, which impedes aerial-ground matching.

*5.2.3 Comparison with Silhouette and Pose-based Methods.* In our comparative analysis of skeleton-based and silhouette-based gait recognition methods. **1)** It is observed that silhouette-based approaches generally outperform skeleton-based methods. This disparity in performance can be attributed to the fact that silhouette segmentation is less impacted by variations in views, because it

 

relies on distinguishing human shapes based on the contrast between the foreground and background colors. Conversely, pose estimation methods, which are fundamental to skeleton-based approaches, are more sensitive to view changes, particularly when the lower body of the human is occluded in aerial views. **2)** Among the skeleton-based models, GPGait [6] demonstrated the best outcomes. This success is likely due to GPGait's Part-Aware Graph Convolutional Network's ability to effectively extract fine-grained local information, showcasing its robustness in skeleton-based gait recognition. **3)** In silhouette-based methods, Vertical Distillation [12] utilizes ground view features as the teacher to guide the learning of aerial view features. However, since the ground view data are not directly incorporated into the training process but only treated as a pre-trained teacher, the model cannot classify the ground view data. The results show that Vertical Distillation achieves results comparable to GaitBase in the Aerial-to-Aerial protocol, but its performance significantly degrades in other settings.

*5.2.4 Comparison of Frame Continuity.* GaitGL [15] processes ordered sequences through 3D CNNs to extract temporal information, while GaitSet [1] takes unordered sets as input and utilizes temporal pooling to acquire set-based information. In the Aerial-to-Aerial protocol, GaitGL surpasses GaitSet on the DroneGait dataset (53.38% compared to 47.87%) but underperforms on AerialGait (69.59% compared to 75.48%). This is probably due to frame discontinuity in the AerialGait dataset, caused by drone movement. In such scenarios, set-based approaches, including GaitSet, GaitBase, and our AGG-Net, demonstrate superior robustness under the condition of discontinuous frames.

**Table 3: Ablation study of each module on AerialGait dataset. Prot. denotes Aerial-Ground Prototype Learning.**

| Base | SOUL | FOUL | Prot. | AerialGait | | | |
|------|------|------|-------|------|------|------|------|
| | | | | A->A | G->G | G->A | A->G |
| ✓ | | | | 81.01 | 96.42 | 72.24 | 75.44 |
| ✓ | ✓ | | | 82.14 | 96.97 | 73.14 | 77.14 |
| ✓ | ✓ | ✓ | | 84.54 | 97.19 | 73.98 | 78.52 |
| ✓ | ✓ | | ✓ | 83.04 | 97.02 | 75.27 | **80.47** |
| ✓ | ✓ | ✓ | ✓ | **84.92** | **97.32** | **76.00** | 80.35 |

### 5.3 Ablation Study

We evaluate the effectiveness of each module on AerialGait, and the results is shown in Table 3. **1)** The incorporation of the SOUL module substantially enhances the model's capacity to focus more discriminative silhouette regions under various views and to adjust to blurring effects due to drone movements, thereby improving performance across all protocols. **2)** The FOUL module projects the features into different Gaussian distributions, significantly encouraging the model to learn a wider distribution based on the learned variance. This further enhances the model's ability when facing gait sequences captured from diverse aerial viewpoints. The third line of Table 3 show that the results have particularly improved, especially in the Aerial-to-Aerial protocol. **3)** The introduction of the Aerial-Ground Prototype Learning module proves effective in aligning

the distributions of aerial and ground views. This module significantly enhances performance, especially in the Aerial-to-Ground and Ground-to-Aerial protocols.

By integrating these modules, our approach effectively extracts the gait features from aerial and ground views and bridges the domain gap between them, achieving a rank-1 accuracy of 84.92%, 97.32%, 76.00%, and 80.35% across the four protocols, respectively. The results demonstrate that all the proposed components contribute consistently to the performance.

### 5.4 Visualization of Feature Distribution

To intuitively demonstrate the effectiveness of our method to bridge the gap between aerial and ground views, we utilize t-SNE [24] to visualize the feature distribution of randomly selected 8 identities on AerialGait. As shown in Figure 6, the feature distribution in the base model exhibits significant domain gap between aerial and ground views, potentially impeding aerial-ground gait recognition as the number of identities increases. In contrast, our AGG-Net successfully aligns the distributions of aerial and ground views, resulting in a more compact and unified feature distribution. Our model alleviate the domain gap between aerial and ground views, enhancing the overall performance in the task of aerial-ground gait recognition.

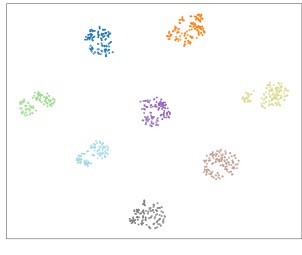 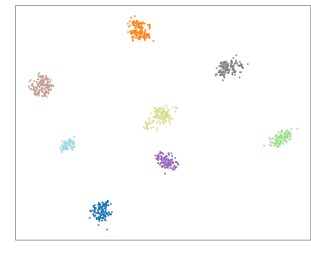

(a)                                    (b)

**Figure 6: The t-SNE [24] visualization of feature distribution in (a) our base model in Section 5.3 and (b) AGG-Net. Different colors represent different identities. Hollow circles ∘ and solid circles • denote features from aerial and ground views, respectively. Best viewed in color and zoomed in.**

### 6 CONCLUSION

In this work, we propose AerialGait for aerial-ground gait recognition. The dataset contains characteristics such as variations in views, motion blur, and other complexities. Comprehensive experiments are conducted on this datasets. Furthermore, we present the Aerial-Ground Gait Network (AGG-Net), which integrates uncertainty learning and prototype learning. AGG-Net effectively bridges the gap between aerial and ground views and demonstrates superior performance on both the AerialGait and DroneGait datasets. In the future, we expect that the AerialGait dataset will encourage more research into the task of aerial-ground gait recognition.

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
