# OpenReview forum: "AerialGait: Bridging Aerial and Ground Views for Gait Recognition"
_acmmm.org/ACMMM/2024/Conference — MM2024 Poster_

### Official Review · Reviewer_HKxH · 2024-05-24

**Rating:** 4
**Confidence:** 3

**Summary:**

This paper introduces AerialGait, a comprehensive dataset designed for aerial-ground gait recognition. The dataset includes 82,454 sequences, totaling over 10 million frames from 533 subjects, captured using both aerial drones and ground surveillance cameras. Data collection spans five diverse surveillance sites, ensuring realistic simulations of various real-world settings.
The authors also propose the Aerial-Ground Gait Network (AGG-Net), which employs uncertainty learning to extract discriminative features from aerial views and uses prototype learning to cluster features across both aerial and ground views.

**Strengths:**

### 1. AerialGait is a highly comprehensive and diverse dataset that simulates real-world  scenarios effectively.
### 2. The dataset introduces new challenges such as motion blur and frame discontinuity, pushing the boundaries of current gait recognition methods.
### 3. The proposed AGG-Net model demonstrates strong performance, addressing the domain gap between aerial and ground views.

**Limitations:**

### 1. The dataset includes various challenges such as different views, low resolution, varying illumination, and motion blur. However, the experiments do not explicitly demonstrate how existing methods and AGG-Net perform under these specific challenges.

### 2. Data Collection and Annotation Details: The paper could benefit from a more detailed description of the data collection and annotation process. Specifically, including statistics on the differences between aerial and ground perspectives would highlight the importance of bridging the gap between these viewpoints.

### 3. In Figure 3, the difference between aerial and ground views is not strongly illustrated. Selecting images that better highlight the distinct characteristics of each viewpoint would improve the clarity and impact of the visual presentation.

**Suitability:**

3

---

### Official Review · Reviewer_ojBA · 2024-05-25

**Rating:** 4
**Confidence:** 2

**Summary:**

This paper presents AerialGait, a comprehensive dataset for aerial-ground gait recognition, and proposes a method which achieves state-of-the-art performance on both the AerialGait and the DroneGait datasets in various experiment settings.

**Strengths:**

1. This paper constructs a large-scale aerial-ground gait recognition dataset named AerialGait for the application of drone-based gait recognition. This dataset has the following unique characteristics: 1) Significant variations in views, resolutions and illumination. 2) Motion blur and frame discontinuity. 3) Domain gap between aerial and ground views.
2. This paper present the Aerial-Ground Gait Network (AGG-Net) and achieve state-of-the-art performance on both the AerialGait and the DroneGait datasets in various experiment settings.
3. The ablation study shows the contribution and efficacy of the individual modules comprising the AGG-Net.

**Limitations:**

About the Method:

1. In Section 4.3, the AGG-Net introduces an Aerial-Ground Prototype Learning module which utilizes a prototype shape of NxDxC, where each identity in the training set corresponds to a prototype feature. Line 606 states that "The prototype can be treated as the most representative examples in the dataset." However, this explanation is somewhat confusing. Could the authors clarify the role and definition of these prototypes within the module? A more detailed explanation would help in understanding how these prototypes contribute to the model's learning process.
2. Lines 662-665 describe how prototype features are used to compute the loss for network training supervision, aiming to minimize the distribution differences between aerial and ground views. Why do the authors choose to use prototype features for this calculation, rather than directly using the aerial and ground view features? An explanation on the choice of prototype features over direct feature comparison for loss calculation would be valuable, particularly in understanding how it benefits the minimization of distribution differences between the two views.

About the dataset:

In addition to the data examples shown in Figure 3, could the authors include some data visualizations with vertical viewing angles over 60°? Will the results of data preprocessing such as human segmentation, parsing and pose estimation be poor in this viewing angle?

About the Experiments:

1. As shown in Figure 2, the results for A->G are higher than G->A on the AerialGait dataset, while the results for G->A are higher than A->G on the DroneGait dataset. Could the authors provide an analysis of this inconsistency in the experimental results under cross-view settings on the two datasets?
2. In Lines 722-726, it is mentioned that “the DroneGait dataset is divided into three subsets, defined by their vertical viewing angles: 0°, 30°-60°, and 60°-80°.” In the experiments, are the 0° subset classified as ground views and the 60°-80° subset as aerial views? If so, was the 30°-60° data not used in the experiment? Clarification on this point would be helpful.
3. Could the authors provide the detailed experimental results for the different vertical viewing angles, such as 30°-60° and 60°-80°? Is there a significant difference in the experimental results across these different vertical view cases?
4. In Lines 731-733, Table 2 shows the average Rank-1 accuracy under conditions of normal walking, carrying bags, and changing clothes. Could the authors provide the detailed Rank-1 accuracy for each of these cases and offer some analysis of these results?

**Suitability:**

2

---

### Official Review · Reviewer_j2Kp · 2024-05-27

**Rating:** 3
**Confidence:** 4

**Summary:**

This paper provide a comprehensive dataset for aerial-ground gait recognition. A comprehensive analysis of existing gait recognition methods are conducted on the collected AerialGait dataset. The paper propose a Aerial-Ground Gait Network (AGG-Net) and achieves state-of-the-art performances.

**Strengths:**

1. The dataset is very comprehensive and has much potential to facilitate the research field.
2. The paper is easy to follow.

**Limitations:**

1. The proposed method is not specially designed to tackle the cross-view problem.
2. The performance is already very promising indicating that the dataset is already tending to saturate. However, the cross view gait recognition is a very challenging task. Maybe the dataset partition is not very suitable, such as  a small gallery size.

**Suitability:**

3

---

### Official Review · Reviewer_GFzA · 2024-05-27

**Rating:** 4
**Confidence:** 3

**Summary:**

This paper presents a new dataset for aerial-ground gait recognition, which has several unique features; 1)significant variations in views, resolutions, and illumination; 2) motion blur and frame discontinuity 3)domain gap between aerial and ground views), and also propose a Aerial-Ground Gait Network, which achieves SOTA on both AerialGait and DroneGait datasets.

**Strengths:**

1. This paper proposes a new dataset for aerial-ground gait recognition, which potentially supplies the research field.

2. The proposed Aerial-Ground Gait Network achieves the SOTA performance.

**Limitations:**

1. The proposed method in this paper can also be applied to traditional gait recognition tasks. Therefore, more experiments are expected to be conducted on other datasets to verify the effectiveness. (i.e. OU-MVLP, CASIA-B, and GREW).

2. The proposed method is complicated with multiple hyperparameters, (e.g. loss weights, The max/min area ratio in SOUL; The updating ratio in Eq.(4)). It's better to explain the setting of these hyperparameters or show parameter analysis in experiments.

3. The movitation can be further explained. Moving the camera will affect the gait information collecting, and self-obscuration in the drone view could also cause the loss of gait information. Why do we need a sky view with a moving camera in Gait Recognition tasks.

**Suitability:**

3

---

### Meta-Review · Area_Chair_rdtt · 2024-07-02

**Recommendation:** Accept (Poster)
**Confidence:** 3

**Metareview:**

This papers propose AerialGait for aerial-ground gait recognition. The dataset contains characteristics such as variations in
views, motion blur, and other complexities. In addition,  the authors present the
Aerial-Ground Gait Network (AGG-Net), which integrates uncertainty learning and prototype learning.